# Development of 3D ZnO-CNT Support Structures Impregnated with Inorganic Salts

**DOI:** 10.3390/membranes12060588

**Published:** 2022-05-31

**Authors:** Stefania Chiriac, Maria-Eliza Puscasu, Ioan Albert Tudor, Alexandru Cristian Matei, Laura Madalina Cursaru, Radu Robert Piticescu

**Affiliations:** National R&D Institute for Non-Ferrous and Rare Metals, INCDMNR-IMNR, 077145 Pantelimon, Romania; schiriac@imnr.ro (S.C.); epuscasu@imnr.ro (M.-E.P.); atudor@imnr.ro (I.A.T.); alex.matei@imnr.ro (A.C.M.)

**Keywords:** zinc oxide-carbon nanotubes composite, 3D printing, porous nanostructured support structure, PCMs inorganic salts, impregnation, solvothermal process

## Abstract

Carbon-based materials are promising candidates for enhancing thermal properties of phase change materials (PCMs) without lowering its energy storage capacity. Nowadays, researchers are trying to find a proper porous structure as PCMs support for thermal energy storage applications. In this context, the main novelty of this paper consists in using a ZnO-CNT-based nanocomposite powder, prepared by an own hydrothermal method at high pressure, to obtain porous 3D printed support structures with embedding capacity of PCMs. The morphology of 3D structures, before and after impregnation with three PCMs inorganic salts (NaNO_3_, KNO_3_ and NaNO_3_:KNO_3_ mixture (1:1 vol% saturated solution) was investigated by scanning electron microscopy coupled with energy-dispersive X-ray spectroscopy (SEM-EDX). For structure impregnated with nitrates mixture, SEM cross-section morphology suggest that the inorganic salts impregnation started into micropores, continuing with the covering of the 3D structure surface and epitaxial growing of micro/nanostructured crystals, which led to reducing the distance between the structural strands. The variation of melting/crystallization points and associated enthalpies of impregnated PCMs and their stability during five repeated thermal cycles were studied by differential scanning calorimetry (DSC) and simultaneous DSC-thermogravimetry (DSC-TGA). From the second heating-cooling cycle, the 3D structures impregnated with NaNO_3_ and NaNO_3_-KNO_3_ mixture are thermally stable.

## 1. Introduction

In recent years, additive manufacturing, also known as 3D printing, has shifted from a tool used exclusively in the industry for rapid prototyping to a new far-reaching approach to the development of high value products. Researchers are also exploring the potential of using 3D printing techniques for composite materials printing [1,2,3,4,5].

On the other hand, its use in fabrication of 3D porous structure as support for phase change materials (PCMs) is relatively new. The use of additive manufacturing techniques could provide more control over the design of these porous structures and provide a new concept for their preparation, additive manufacturing being able to produce porous structures of various shapes, types and designs (regardless of its complexity) that cannot be made using conventional techniques, such as the phase inversion method or sintering [6,7,8,9,10].

Recently, 3D printing technology has been used to fabricate and embed various architectures with PCMs. PCMs are substances which absorb or release energy during phase transition, they are highly used in the building sector for heating/cooling purposes as well as in the thermal energy storage sector [11,12,13,14,15,16,17,18,19]. 

Depending on their nature, PCM can be organic, inorganic or metal alloy compounds, used at low (<100 °C), medium (100–200 °C) and high (>200 °C) temperatures [18]. Organic PCMs are usually classified as paraffin (saturated hydrocarbons) or non-paraffin compounds (fatty acids, alcohols, glycols, esters). Inorganic PCMs can be salt hydrates (nitrates, fluorides, chlorides, sulphates, hydroxides, carbonates, etc.) or metallic alloys (Al-Si, Al-Si-Mg, Al-Si-Cu, Al-Mg-Zn, etc.). There are also various types of eutectic PCMs: inorganic-inorganic, inorganic-organic or organic-organic materials [11,13,15]. Low temperature applications (<100 °C) of PCMs for thermal energy storage systems include passive heat/cold buffering in food industry, insulation building materials, domestic space heating, etc. [18]. Some examples of high temperature applications (>100 °C) are solar power generation system or industrial waste heat recovery [18]. Thermal energy can be stored by physical processes (sensible heat and latent heat) and chemical processes (thermo-chemical heat). Among these methods, latent heat thermal energy storage (LHTES) system using phase change materials is the most efficient due to their high energy density per unit mass and per unit volume, and ability to provide heat at a constant temperature [11,12,13,14,15,16,17,18,19]. 

Due to their high latent heat of fusion and thermal stability, organic PCMs seem good candidates for LHTES systems [20,21]. However, due to their low thermal conductivity, they were impregnated into 3D printed metallic structures [22,23]. 

Inorganic PCMs can store and release energy in the solid phase through different processes such as crystallographic structure transformations or changes from amorphous to crystalline structure. They have better thermal conductivity but can be corrosive and have lower heat transfer properties compared to other storage media [22,24,25,26,27,28]. 

Therefore, PCMs must be encapsulated in shells or impregnated in a porous structure, thus avoiding contact with the environment. A developing method for increasing the heat transfer of inorganic PCM is to micro-encapsulate PCM in various organic and inorganic materials, usually by sol-gel processes leading to an increase in the exchange surface between PCM and the transfer fluid. 

Impregnation technique is an important method of improving energy storage ability of PCMs [29,30,31,32,33]. Inorganic salt (PCM material) absorbs or releases the latent heat during the melting or recrystallization cycle, while the porous structure in which is impregnated (embedded) prevents PCM from leaking during melting [13,25,27,29,34,35,36]. Several materials have been used as porous support structures for PCMs [22,30], such as metal foams, vermiculite [37], perlite [38], bentonite [39], kaolin [40,41], diatomite [42,43]. 

Rathore and Shukla [37] have studied a porous structure based on Expanded Graphite (EG) and Expanded Vermiculite, prepared through physical blending followed by vacuum impregnation, aiming to improve thermophysical properties of a commercial organic PCM. In 2017, Karaipekli et al. [38] enhanced thermal energy storage properties of expanded perlite (ExP)/n-eicosane (C20) composite by using CNT - known for their good thermal conductivity. In other study, bentonite-based PCMs composite was produced by impregnation of capric acid, polyethylene glycol, dodecanol and heptadecane into bentonite clay [39]. An increase of thermal conductivity has been observed by addition of 5 wt% EG. Jafaripour et al. [41] have successfully impregnated stearic acid (a fatty acid) into the pores and layers of kaolin as the supporting material by melt impregnation process. In 2022, Ren et al. [42] developed a new shape-stabilized PCM (SSPCMs) by impregnating hydrophobic modified diatomite with ternary fatty acids (lauric acid, myristic acid and palmitic acid) and incorporating SSPCMs into mortar. 

Yet, the poor interaction between PCMs and the porous structure cannot prevent leakage when using large amounts of PCM [30]. Currently, researchers’ investigations are directed toward finding the proper porous structure as PCM support. It has been found that the pore size, surface properties and the structure of the materials used as support have a major influence on the thermal properties of PCMs.

Carbon-based materials are promising candidates for enhancing thermal properties of (PCMs) without lowering its energy storage capacity applications due to their high thermal conductivity, large specific surface area, good compatibility with the PCMs, low density and lightweight [44,45,46,47].

New structures based on porous graphene foam, EG and carbon nanotubes (CNTs) hybrid material impregnated with PCMs have been studied for thermal energy storage [48,49,50,51,52]. 

Commonly, PCMs are embedded into the porous structure of graphite by infiltration or differently, carbon-based particles are dispersed into melted PCMs, preparing graphite-based/PCM composites when solidified. The thermal conductivity of these composites is greatly enhanced but the main limitation is the poor homogeneity of the dispersion of conductive particles into the PCM [53,54]. 

It has been found recently that incorporation of Ag doped ZnO nanomaterials within organic eutectic mixture enhances the thermal stability and thermal conductivity of microencapsulated PCMs [19]. It is known that ZnO nanomaterials have high thermal conductivity, excellent thermal stability, and antibacterial activity (non-toxicity), thus being capable to enrich the functional diversity of composite PCMs [19,55]. Wang et al. [56] has studied ZnO/EG as supporting material for palmitic acid (PA) as phase change material for heat storage. The hybrid structure of ZnO/EG can prevent the agglomeration of ZnO and provide high specific surface area and pore volume to adsorb PA The thermal conductivity of the PA/ZnO/EG-6% composite PCMs was increased by 137.5% compared with pure PA [56].

To increase the heat transfer properties of inorganic salts and the stability to thermal cycles, this paper aims to develop ZnO-CNT 3D support structures made by additive manufacturing, impregnated with inorganic salts such as NaNO_3_ and KNO_3_ (as PCMs) by solvothermal processes at high pressures (1000–3000 bar). NaNO_3_ and KNO_3_ are among the most widely used inorganic PCMs, with melting points suitable for thermal energy storage applications. Moreover, combination of these salts presents advantages such as high energy density, high heat capacity, low cost, thermal stability [57].

Three PCM formulations were tested in this study: NaNO_3_, KNO_3_ and NaNO_3_:KNO_3_ 1:1 vol% mixture. Thermal stability and embedding capacity of 3D structures impregnated with the three types of PCMs were compared to establish the optimum variant.

The use of nanostructured ZnO aims to enhance the thermal transfer due to the formation of flower-like structures with high specific surface and good chemical compatibility with CNT because of negative charges of hydroxylated zinc species formed on the surface in hydrothermal conditions, as previously demonstrated in [58]. 

It is known that zinc oxide nano-flakes grown on the surface of carbon nanofibers have potential applications in developing energy storage devices (hybrid supercapacitors) due to synergistic effect of ZnO and carbon nanofibers [59,60].

As a novelty, ZnO-CNT-based powder prepared through a hydrothermal process at high pressure was used in this paper to obtain porous 3D structures with embedding capacity of PCMs (NaNO_3_, KNO_3_ and NaNO_3_ - KNO_3_ mixtures). The impregnation of ZnO-CNT 3D structures with PCMs was performed by solvothermal processes at high pressures (1000–3000 bar) in nitrate solutions, selecting the most suitable inorganic salt to be impregnated. The variation of melting/crystallization points and enthalpies of impregnated PCMs and their stability during five thermal cycles were studied by differential scanning calorimetry (DSC) and simultaneous DSC-thermogravimetry (DSC-TGA).

## 2. Materials and Methods

### 2.1. Hydrothermal Synthesis of ZnO-CNT Composite Nanopowder

ZnO-CNT *composite* nanopowder with a mass ratio of CNT: ZnO of 1:10 have been synthesized by hydrothermal method at 200 °C, as described in [56].

### 2.2. Preparation of ZnO-CNT Nanopowders-Based Pastes for 3D Printing

The pastes were prepared starting from ZnO-CNT composite nanopowders and sodium salt of poly (acrylic acid), as a binder. The two components were mixed during 2–5 min at 2000 rpm and defoamed 4 min at 2000 rpm in a Thinky-ARE 250 centrifugal planetary mixer (Thinky Corporation, Tokyo, Japan) to homogenize the ZnO-CNT based pastes and remove air bubbles. These homogeneous pastes were further used for the additive manufacturing of 3D structures with the 3D-Bioplotter EnvisionTEC Starter System (EnvisionTEC GmbH, Gladbeck, Germany).

### 2.3. Additive Manufacturing of 3D Structures Based on ZnO-CNT Pastes 

3D structures based on the pastes prepared from ZnO-CNT nanopowders were obtained by robocasting technique (an extrusion-based 3D printing technique) using the 3D-Bioplotter System. The working parameters were cube-like structure with 10 × 10 × 5 mm^3^ in size, 1.3 mm strand distance, nozzle diameter of 400µm, angle between layers of 90° and the printing speed of 9–10 mm/s. More details about 3D printing process of ZnO-CNT structures are presented in [61].

### 2.4. Post-Printing Treatment of ZnO-CNT 3D Support Structures 

To improve the properties and mechanical resistance of the 3D structures used for impregnation at high pressures (1000–3000 bar) in solvothermal conditions, the obtained 3D objects were heat treated in air at controlled heating from room temperature to 490 °C, using a Carbolite Gero (30–3000 °C) Verder Scientific furnace with digital programmer. 

Subsequently, the resulted 3D structures were impregnated with inorganic salts - PCMs, by solvothermal process at high pressures.

### 2.5. Impregnation of ZnO-CNT 3D Structures with Inorganic Salts (PCMs)

Impregnation of PCM salts was performed by solvothermal processes in high pressure conditions (1000–3000 bar), using HP Systems autoclave (ALTIFORT HP SYSTEMS, Perigny, France). This process is described in Figure 1. Impregnation takes place under isostatic conditions, after the 3D printing, the structures have been placed inside a plastic bag filled with inorganic salt solution. This bag was sealed and introduced in the autoclave vessel, being kept at 100 °C for 30 min. The pressure is transmitted to the wet bag equally in all directions, thus favoring the embedding of inorganic salts into 3D structures.

Three different PCM solutions were tested: (1) sodium nitrate (90g NaNO_3_/100 mL H_2_O); (2) potassium nitrate (90g KNO_3_/100 mL H_2_O); (3) NaNO_3_:KNO_3_ = 1:1 (vol%). 

Resulting 3D samples impregnated with inorganic salts were dried at room temperature and subjected to morphological characterization and thermal analysis to investigate their surface and thermal properties.

To demonstrate the embedding capacity of inorganic salts as well as their thermal stability in performing PCMs properties, the impregnated 3D structures were characterized by morphological (SEM/EDX techniques) and thermal methods, performing repeated thermal cycles by (DSC) and thermogravimetry (TGA-DTG). 

### 2.6. Characterization Methods 

#### 2.6.1. Morpho-Structural Characterization 

The morphology of 3D structures, before and after PCMs-impregnation, was studied by scanning electron microscopy (SEM) and energy-dispersive X-ray spectroscopy (EDX) with a FEI Quanta 250 electron microscope (FEI, Eindhoven, The Netherlands) in Low Vacuum mode, using the secondary electron detector, and BSE mode respectively. 3D structures have also been characterized by X-ray diffraction (XRD). The samples subjected to X-ray diffraction were ground in a agate pestle mortar. The data acquisition was performed on the BRUKER D8 ADVANCE diffractometer using the DIFFRACplus XRD Commander software (Bruker AXS), by the Bragg-Brentano diffraction method, Θ—Θ coupled in vertical configuration, with the following parameters: CuKa radiation, 2θ region 28–74° 2θ step 0.03°. The data processing was done with the help of the DIFFRAC.EVA release 2019 program from the DIFFRAC.SUITE.EVA software package and the ICDD PDF4 + 2022 database. 

Mean crystallite sizes were calculated using Debye-Scherrer formula [61]: $$D=\frac{k\lambda }{\beta cos}$$
where:*D*—the average crystallite size, in nm*β*—the line broadening at half the maximum intensity, in radians,*λ*—the X-ray wavelength, in Å;*k*—constant; *k* = 0.9 according to Bragg or 0.70 < *k* < 1.70 according to Klug and Alexander*θ*—diffraction (Bragg) angle 

Percentage composition of the material was determined using TOPAS software.

#### 2.6.2. Thermal Characterization 

ZnO-CNT 3D structures were characterized by DSC method using a DSC F3 Maia (Netzsch, Selb, Germany) equipment and DSC-TGA analysis using the Setaram Setsys Evolution device (Setaram Instrumentation, Caluire, France), in argon atmosphere. 

DSC measurements were performed in perforated aluminum crucibles and heated up to 350 °C, with heating and cooling rate of 10 K/min. Experimental data processing was performed using Proteus Analysis software (Netzsch, Selb, Germany). In the case of DSC-TGA analysis, 3D samples were introduced in alumina crucibles and heated up to 350 °C, with heating and cooling rate of 10 K/min. Experimental data were processed with Calisto software v1.097 (Setaram Instrumentation, Caluire, France). Samples impregnated with inorganic salts were subjected to 5 successive heating-cooling cycles in argon gas, in the temperature range 20–350 °C, by DSC and DSC-TGA methods, respectively.

The upper limit of 350 °C has been set knowing that boiling point is 380 °C for sodium nitrate and 400 °C for potassium nitrate.

## 3. Results and Discussion 

### 3.1. Morphostructural Characterization of ZnO-CNT Composite-Based 3D Support Structures 

Digital images of ZnO-CNT 3D structures, obtained by the robocasting technique, to be used as porous support material for inorganic salts impregnation, are presented in Figure 2. The structures have 1 × 1 cm square geometry with a 3D multilayered structure pattern having 90 degrees rotation angle between the two successive strands layers deposition.

Figure 3 shows the SEM micrographs of the obtained 3D support structure after calcination (for mechanical stability) and before impregnation. One can observe that the strand width of the non-impregnated 3D structures varies in the range of 354–391 µm, in accordance with the nozzle diameter of 400 µm (Figure 3a). The observed contraction (3–10%) is due to the elimination of volatile organic products during the heat treatment of the as-printed 3D structure. A smaller distance between strands of 1.17–1.19 mm, when compared with the initially set value (1.3 mm), can be observed, also due to material contraction during the post-printing heat treatment.

Homogeneous, cracks-free and smooth surface aspect can be observed from Figure 3a, while the nanoscale morphology observed in Figure 3b reveals porous aggregates of ZnO 1D nanorod, which diameter ranges between 190–300 nm and being of 1.1–1.2 μm in length. Due to both very small sizes (10 nm outer diameter, 4.5 nm inner diameter) and the very small quantity (1 wt% in nanocomposite powder and 5% into the 3D structure), CNT cannot be observed in these images. Cross section images of ZnO-CNT-based 3D support structure reveals a porous structure inside strands (Figure 3c,d). 

X-ray spectrum of ZnO-CNT-based 3D support structure before impregnation is depicted in Figure 4. The diffraction peaks located at 31.76°, 34.41°, 36.24°, 47.53° and 56.58° show the presence of hexagonal wurtzite ZnO [62,63] as major crystalline phase, identified as zincite (PDF Card no. 00-036-1451). Lattice constants are 𝑎 = 𝑏 = 0.325 nm and 𝑐 = 0.5207 nm. Crystallite size of ZnO on (101) direction is 96.98 nm. The existence of CNT cannot be observed, its concentration in the final 3D structure being below the detection limit of the diffractometer. 

### 3.2. Morphostructural Characterization of PCMs-Impregnated 3D Structures

Figure 5 shows the SEM micrographs of the obtained 3D support structures after impregnation with the three above mentioned inorganic salts. One can observe that the presence of NaNO_3_ salt impregnated in the 3D structure led to an inhomogeneous aspect, (Figure 5a) due to mostly inhomogeneous salt deposition on the surface, which increase the strand width to 401–448 µm. So, the distance between the strand varies, from that similar to the one of non-impregnated 3D structure (1.17–1.18 mm), but there are also some regions where this distance is the same as the set value (1.31 mm). Agglomerations of salt nanocrystals can be observed in certain areas (Figure 5b).

In the case of 3D structures impregnated with KNO_3_ (Figure 5c), very smooth, homogeneous, and compact aspect, with very small size crystallite (Figure 5d), of salts layer deposed on the 3D structure surface can be observed. Therefore, the width of the strands varies very little (410–420 µm), while the distance remains constant (1.32–1.34 mm) and closed to that of the non-impregnated structures. Nanowires, conical formations and microbands grown in the space between the stands can be observed in the detail from Figure 5d.

In the case of NaNO_3_-KNO_3_ mixture (Figure 5e,f and Figure 6e,f), one can observe the most interesting morphology, including the presence of well-developed and defined 2D micro/nanostructured salts crystals grown inside the space between the perpendicularly deposed strands of the 3D multilayered structure. For this sample, the detailed SEM images (Figure 5 and Figure 6) suggests that the inorganic salts impregnation started into micropores, continue with the covering of the 3D structure surface and finally with the epitaxial growing of micro/nanostructured crystals, using as seeds the crystallites of the previous deposed layer. Crystallization germs on the strand’s surface led to the penetration of the growing 2D nanocrystals into the 3D structure gaps space (Figure 6).

The EDS spectra depicted in Appendix A (Appendix A) identified the existence of inorganic salts on the surface of ZnO-CNT materials through the presence of Na (Appendix A) and K (Appendix A) and Na+K (Appendix A) elements. The cracks observed, especially in Figure 5e are most probably due to high isostatic pressure applied during the impregnation, but also due to the penetration of a high amount of high size of 2D micro/nanostructured salt crystal impregnated from solution.

Figure 7 shows XRD patterns of ZnO-CNT-based 3D support structure impregnated with PCM salts.

The XRD spectra of the PCM impregnated samples shows the presence of hexagonal wurtzite ZnO as major crystalline phase (zinc oxide, PDF Card no. 04-015-4060 in the case of NaNO_3_ impregnated sample and zincite, PDF Card no. 00-036-1451 for KNO_3_ and NaNO_3_-KNO_3_ impregnated samples). The presence of crystalline rhombohedral NaNO_3_ in the case of NaNO_3_ impregnated 3D structure (Figure 7a) is demonstrated by the characteristic X-ray lines of nitratine (PDF card no. 00-007-0271). The diffraction peaks at 2θ° angles of 22.82°, 29.37°, 31.87°, 35.36°, and 38.94° correspond to the reflection from (012), (104), (006), (110), (113) crystal planes of the rhombohedral nitratine structure [64,65]. Rhombohedral (or trigonal system) is sometimes considered a subdivision of the hexagonal system.

Small intensity peaks of orthorhombic KNO_3_ crystalline phase, identified as niter (PDF Card no. 01-071-1558), according to [66,67] are barely visible in Figure 7b for KNO_3_ impregnated 3D sample and correspond to 2θ° angles of 23.55°, 23.81°, 29.43°, 41.15° and 41.78°. This could be because only a limited amount of salt remained into 3D structure after the solvothermal process, as can be seen at SEM results (Figure 5c and Figure 6d). Instead, KNO_3_ can be observed in Figure 7c (representing NaNO_3_-KNO_3_ mixture impregnated structure) at 2θ° angles of 23.55°, 23.82°, 29.41°, 41.16° and 41.81°, corresponding to crystal planes of niter. The presence of NaNO_3_ in this mixture impregnated sample could be masked by KNO_3_ diffraction peak at 2θ = 29.45° and ZnO diffraction peak at 2θ = 31.8°.

Lattice constant and mean crystallite sizes calculated using Debye-Scherrer formula are presented in Table 1 and Table 2.

According to the data presented in Table 1, the nature of the salt used for impregnation does not significantly affect the lattice constant values, so it is not possible to talk about an insertion of salt ions in the ZnO network of the support structure. At the same time, however, a strong effect is observed on the values of the ZnO crystallite size, which increase from 96.98 nm (in the case of the non-impregnated structure) to 101.19 nm in the case of NaNO_3_ impregnation and decrease dras-tically to 69.95 and 55.64. nm in the case of impregnation with KNO_3_ and NaNO_3_-KNO_3_ mixture, respectively. Corroborating with the morphological aspect presented in the SEM cross-section images (Figure 6), it is possible to suggest a penetration of the pores and intergranular spaces of the support structure, much more accentuated when using the NaNO_3_-KNO_3_ salt mixture, which leads (at high pressures used) to a restructuring of the ZnO powder from the component of the 3D support structure.

### 3.3. Thermal Stability of the as-Fabricated 3D Support Structures

Thermal properties of 3D structures before impregnation with PCM were studied by DSC and simultaneous DSC-TGA methods in an argon atmosphere. Figure 8 and Table 3 show the results of the simultaneous DSC-TGA analysis of the 3D support structure before impregnation with PCMs. The peak temperature values and associated thermal effects resulting from DSC analysis were correlated with weight loss and derivative weight loss curves (Figure 8a). Two important processes with mass loss (Figure 8a) and endothermic effects (Figure 8b) can be observed in the case of the first heating cycle of printed 3D support structures with a total mass loss of ~1.4% with maximum rates at 55 and 240°C, respectively. While the first process can be assigned with solvent/water losses, the second one can be associated with the multi-stage transformation/decomposition of the CNT carboxyl functional groups (195–240 °C) [61]. The heat flow curve (Figure 8b) confirms two main steps decomposition process in the interval of 160–240 °C. 

During the 2nd–5th heating cycles, a slight increase in mass (<0.2%) is observed (detail in Figure 8a). This effect could be caused by gas (Ar) reversible adsorbing (see a reversible mass loss in the cooling cycles) into the 3D structure. These very small (around 0.2%) reversible mass loss/gain, which is more than 6 times lower than weight loss during the first heating cycle, confirmed the stability of the impregnated structures as functional PCMs working system.

Figure 9 shows the five heating-cooling cycles for a non-impregnated 3D ZnO-CNT structure. As was revealed by the DSC-TGA techniques, significant differences between the first and the next four heating cycles can be observed. The characteristic endothermic peaks of this hybrid material at about 240 °C, assigned to structural transformations of CNTs carboxyl functional groups of 3D structure [61] no longer appear in subsequent cycles. In the curves corresponding to cycles two to five, no irreversible processes were observed (Figure 9b).

### 3.4. Thermal Properties and Stability of 3D Structures Impregnated with PCM

Figure 10 shows the DSC-TGA thermogravimetric curves of the 3D support structure after impregnation with NaNO_3_ salt. When comparing with the similar curves of the non-impregnated structure, some differences can be noticed associated with the presence of inorganic salt, i.e., a new mass loss endothermic peak at ~120 °C and a shoulder between 270–290 °C. An increased of total mass loss (1.6 %) during the first heating cycles of the impregnated 3D structure, when comparing with the non-impregnated one (1.376%). 

Thermogravimetric analysis results for PCM impregnated structures are presented in Table 4.

The synthetic DSC curves of the 3D ZnO-CNT structure impregnated with NaNO_3_, KNO_3_ and NaNO_3_-KNO_3_, for five successive heating-cooling thermal cycles, are presented in Appendix A, respectively (Appendix A). The peaks associated with the main processes observed during the five thermal cycles of 3D structures impregnated with the three mentioned inorganic PCMs are summarized in Table 5. The endothermic peak (in heating cycles curves) and exothermic peaks (in cooling cycling curves) correspond to the melting and recrystallization points, respectively (Table 5). 

Consistent and relevant changes in melting and recrystallization temperature were obtained when the two salts were impregnated simultaneously (as a mixture) in the 3D composite structure. It should be noticed that the new melting/crystallization point values are higher than the values for each salt impregnated separately and, unlike in the latter cases, the values remain practically constant during all five cycles. Thus, for the first heating-cooling cycle, while the presence of separated NaNO_3_ and KNO_3_ salts is highlighted by the peaks at 310.3 °C and 327.6 °C, respectively (in heating) and those at 295.3 °C and 292.6 °C, respectively (in cooling), the presence of NaNO_3_-KNO_3_ mixture is highlighted by higher value peaks of 335.8 °C and 330 °C, respectively.

Figure 11 shows the 5 heating-cooling cycles for NaNO_3_, KNO_3_ and NaNO_3_-KNO_3_- impregnated 3D structures.

The DSC curve corresponding to the first heating cycle of the 3D structure impregnated with NaNO_3_ (Figure 11a) shows three more important endothermic processes, one at 240 °C, another of lower intensity at 270 °C and the most intense at 310 °C. Starting with the second heating cycle, the peak at 240 °C disappears confirming an irreversible process of CNTs functional groups decomposition, while the other two indicate reversible processes. The last of these processes (salt melting) is shifted to a lower temperature (306 °C in cycle 2 and 304.5 °C in the last three cycles. These values, as well as the recrystallization temperature (295.3(6) °C) on cooling (Figure 11b and Table 5) confirm the thermal stability of the structure impregnated with NaNO_3_ and are in accordance with the literature data [68]. The melting enthalpy decreases from 58.03 J/g (for the first heating cycle) to 56.09 J/g (for the last heating cycle), due to an increase in the heat flux demonstrating that NaNO_3_ is impregnated into the 3D structure [57,69,70]. Recrystallization enthalpy is around −58 ÷ −59 J/g (Appendix A).

When the impregnation PCM agent was KNO_3,_ (Figure 11c) the endothermic peak assigned to the melting salt process in the first heating-cooling cycle was observed at 327.6 °C, followed by recrystallization at 292.6 °C. The melting point decreases at 320°C after 5 thermal cycles while recrystallization occurs at around 293 °C (Figure 11d). Small melting and recrystallization enthalpies (1 ÷ 3 J/g and −2.5 ÷ −2.6 J/g, respectively) (Appendix A) may suggest that KNO_3_ impregnation of ZnO-CNT 3D structures is less efficient than NaNO_3_. The endothermic peak occurring at 138–142 °C is explained by polymorphic transformations of KNO_3_ [69].

The 3D structure impregnated with 1:1 vol% saturated solution of sodium nitrate and potassium nitrate (Figure 11e), clearly displays, for the first heating cycle, both the characteristic DSC peaks of the ZnO-CNT material (structural transformations and decomposition of functional groups of CNT at ~240 °C) and the melting peak of nitrate salts. An increased melting point of the potassium nitrate impregnated into the 3D structure of ZnO-CNT in the presence of sodium nitrate is demonstrated by the shift of the endothermic peak at 334–335 °C (in each heating cycle) [61]. From the second heating cycle onwards, only salts melting (Figure 11e) or recrystallization (Figure 11f) are observed. The endothermic peak at 134–140°C corresponding to KNO_3_ polymorphic transformations is also observed in all heating cycles (Figure 11e). The effect of mixing impregnation salts (NaNO_3_-KNO_3_) on the thermal stability of impregnated 3D structure is also demonstrated by increasing the recrystallization point at about 330–330.7 °C [70], for all five cooling cycles (Figure 11f). The melting enthalpy of KNO_3_ is about 9 J/g for the first heating cycle, and then rapidly decreases at 0.7 J/g. During cooling cycles, the KNO_3_ recrystallization enthalpy is smaller (−1 J/g) than that of NaNO_3_ (around −13 J/g). Corresponding melting enthalpy of 12 J/g observed for the endothermic peaks positioned at 320–321 °C (Appendix A) is associated with the presence of sodium nitrate and potassium nitrate mixture (Figure 11e and Table 5). 

Our results demonstrated an improved impregnation process, better crystallization of the salt used for impregnation, and better stability for the impregnated structure with the mixture of NaNO_3_ and KNO_3_ salts.

## 4. Conclusions

As a novelty, 3D structures based on hybrid ZnO-CNT 1D nanostructured (nanorods) powder, obtained by the hydrothermal method, were investigated for 3D printing of porous structures as support for impregnation with inorganic salts (as phase change materials). NaNO_3_, KNO_3_ and 1:1 vol% saturated solution of nitrates mixture (NaNO_3_:KNO_3_) were investigated as PCMs impregnation agents, using a solvothermal process in isostatic pressure conditions.

Especially for structures impregnated with nitrates mixture, the SEM cross-section morphology suggest that the inorganic salt impregnation started into micropores, continuing with the covering of the 3D structure surface and finally with the epitaxial growing of micro/nanocrystals. 2D micro/nanostructured PCMs phase has grown inside the space between the perpendicularly deposed strands of the 3D multi-layered structure when the mixt nitrate (NaNO_3_-KNO_3_) solution was used for impregnation.

DSC and simultaneous DSC-TGA techniques have been used to investigate the variation of melting/crystallization points, associated enthalpies of impregnated PCMs and their stability during five heating-cooling cycles (between room temperature, RT and 350 °C). 

Relevant changes in melting and recrystallization temperature were obtained when the NaNO_3_-KNO_3_ salts were impregnated simultaneously (as a mixture) in the 3D composite structure. The melting point (320–321 °C) and corresponding enthalpy values (12 J/g) suggest a synergic effect concerning the thermal stability and PCM activity properties of the 3D ZnO-CNTs structure impregnated with nitrate mixt solution. 

Based on these encouraging results, future studies will be developed to optimize the impregnation pressure value in order to avoid the appearance of cracks in the impregnated 3D structures, but also to tune the morphology and size of the PCM nanocrystals. The composition of the salt solution and the impregnation span represent other important parameters in this respect.

## Figures and Tables

**Figure 1 membranes-12-00588-f001:**
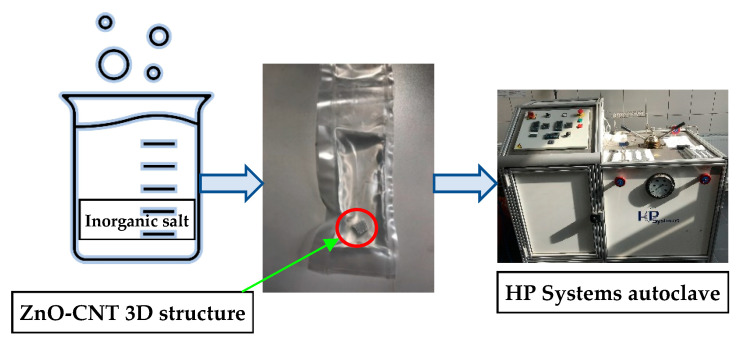
Impregnation of ZnO-CNT composite 3D structures with inorganic salts in isostatic pressure conditions.

**Figure 2 membranes-12-00588-f002:**
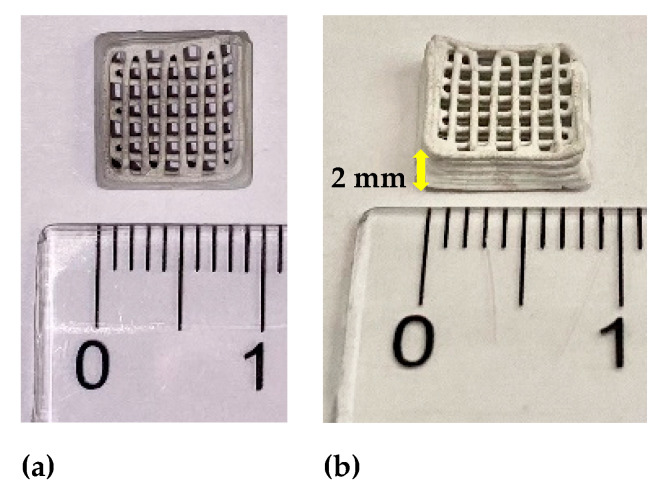
3D structures based on ZnO-CNT composite manufactured by robocasting: (**a**) top view; (**b**) side-view.

**Figure 3 membranes-12-00588-f003:**
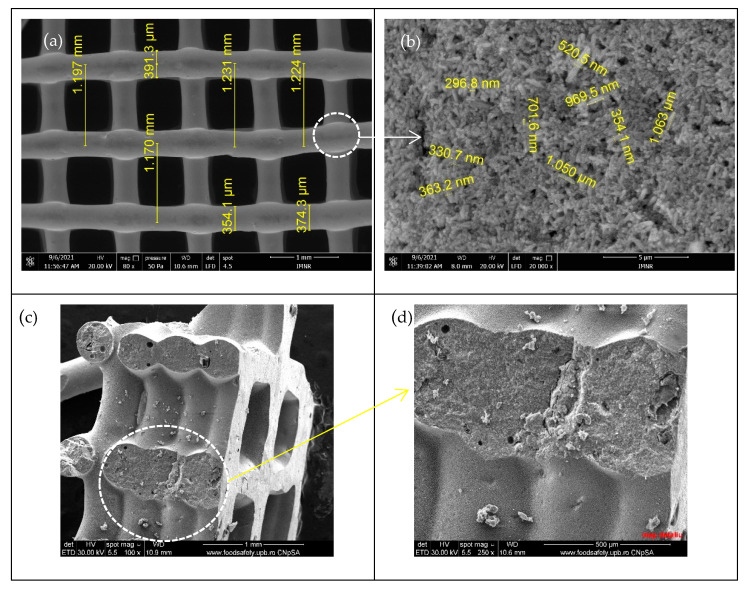
SEM micrographs of ZnO-CNT-based 3D support structure before impregnation: (**a**) 3D structure; (**b**) detail from (**a**) strand morphology; cross-section images: (**c**) 100× magnification; (**d**) 250× magnification.

**Figure 4 membranes-12-00588-f004:**
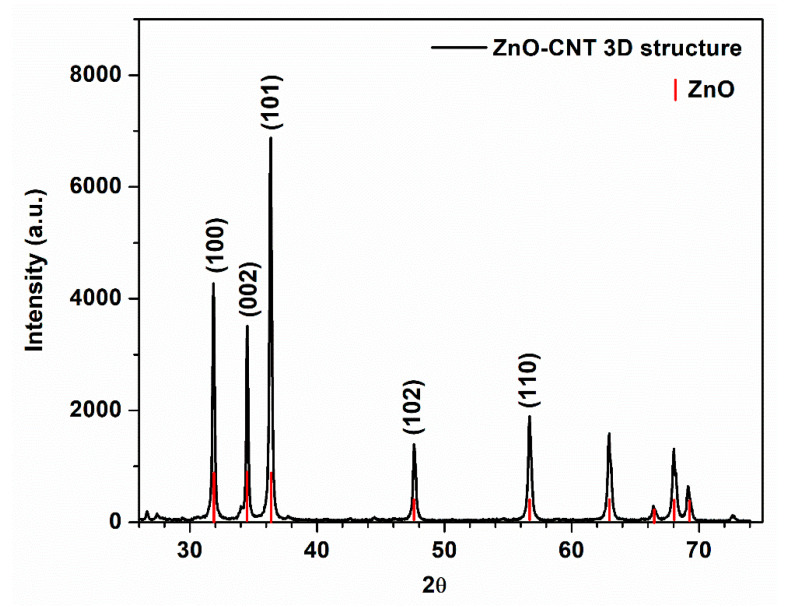
XRD pattern of ZnO-CNT-based 3D support structure before impregnation.

**Figure 5 membranes-12-00588-f005:**
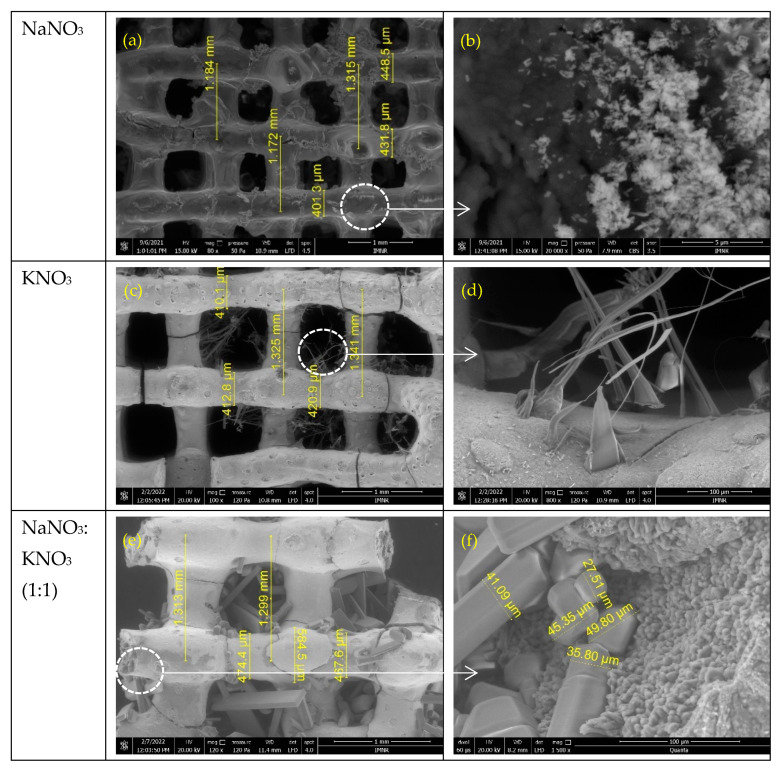
SEM micrographs of ZnO-CNT-based 3D support structure impregnated with PCMs salts: (**a**,**b**) NaNO_3,_ (**c**,**d**) KNO_3_ and (**e**,**f**) NaNO_3_:KNO_3_.

**Figure 6 membranes-12-00588-f006:**
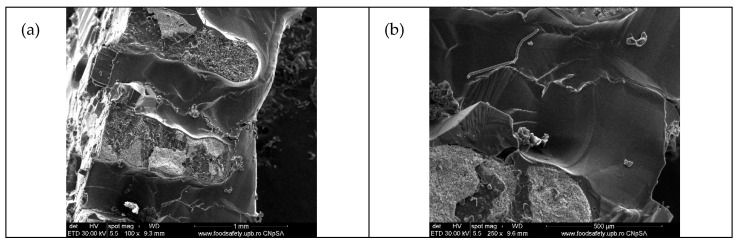
SEM cross-section images of ZnO-CNT-based 3D support structure impregnated with PCMs salts: (**a**,**b**) NaNO_3,_ (**c**,**d**) KNO_3_ and (**e**,**f**) NaNO_3_:KNO_3_.

**Figure 7 membranes-12-00588-f007:**
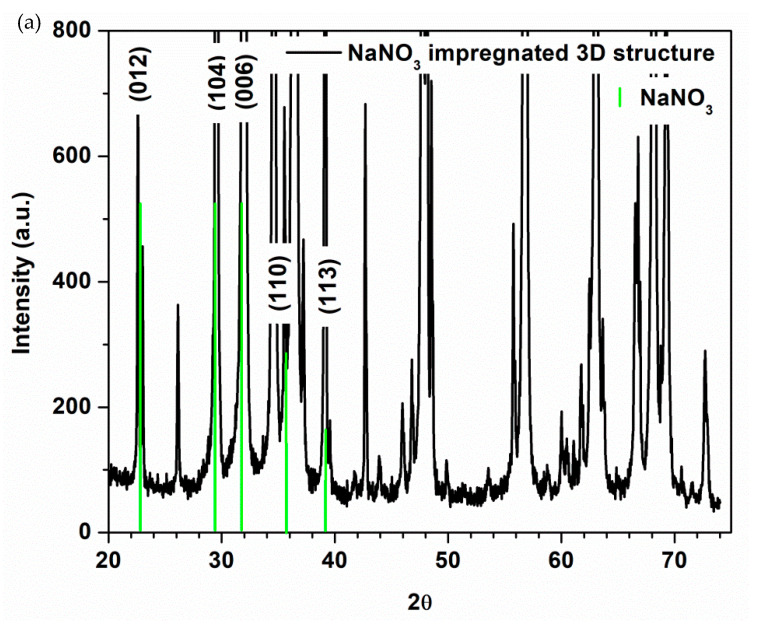
XRD patterns of ZnO-CNT-based 3D support structure impregnated with PCM salts: (**a**) NaNO_3,_ (**b**) KNO_3_ and (**c**) NaNO_3_:KNO_3_.

**Figure 8 membranes-12-00588-f008:**
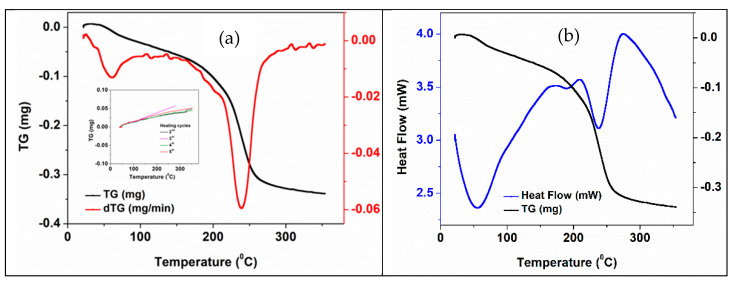
Thermal curves of the non-impregnated 3D ZnO-CNT structure: (**a**) TGA-DTG curves-first heating cycle. The inset shows the TG curves for 2nd–5th heating cycles; (**b**) DSC-TGA curves-first heating cycle.

**Figure 9 membranes-12-00588-f009:**
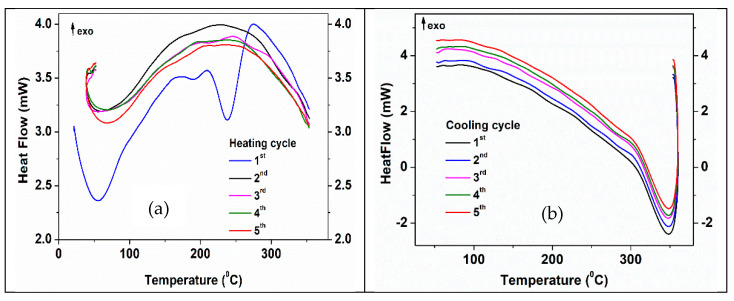
DSC graph of a non-impregnated 3D ZnO-CNT structure: (**a**) 5 heating cycles; (**b**) 5 cooling cycles.

**Figure 10 membranes-12-00588-f010:**
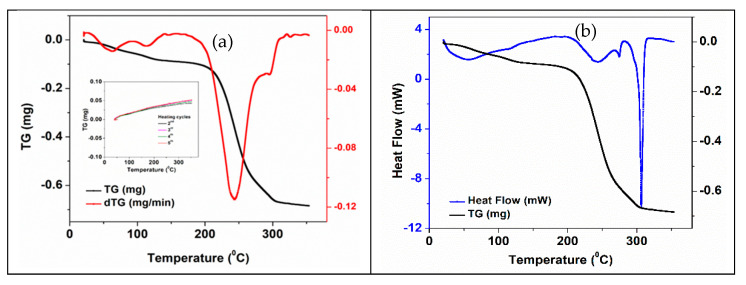
Thermal curves of the NaNO_3_ -impregnated 3D ZnO-CNT structure: (**a**) TG-DTG—first heating cycle. The inset shows the TG curves—2nd–5th heating cycles; (**b**) TG-DSC—first heating cycle.

**Figure 11 membranes-12-00588-f011:**
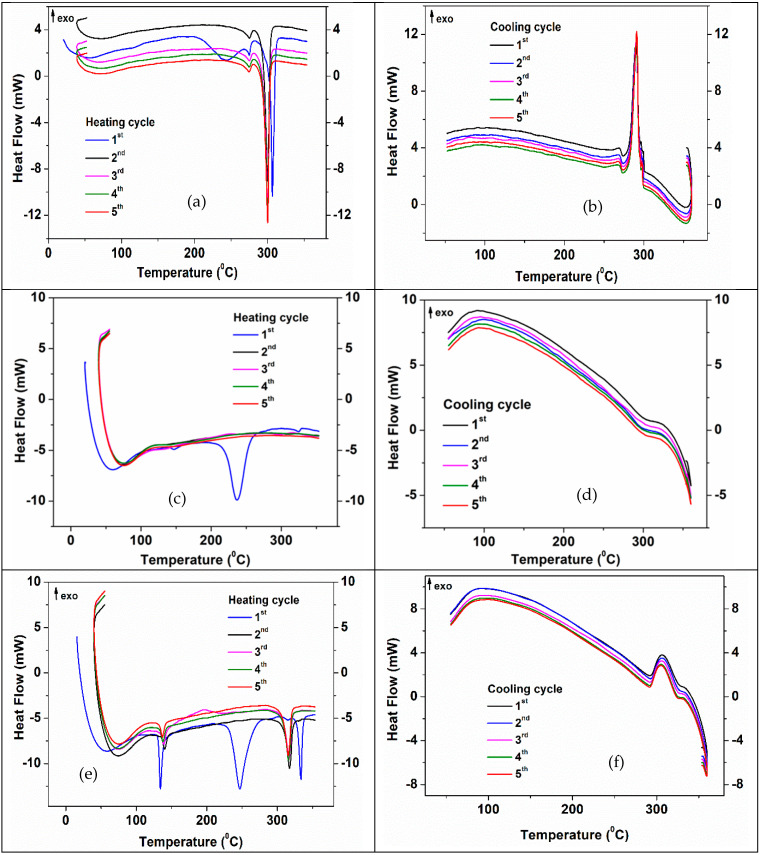
DSC curves for five heating-cooling cycles of the 3D ZnO-CNT structure impregnated with (**a**,**b**) NaNO_3_; (**c**,**d**) KNO_3_; and (**e**,**f**) NaNO_3_:KNO_3_.

**Table 1 membranes-12-00588-t001:** Results of the XRD analysis concerning the effect of impregnation on the structural parameters of ZnO phase of 3D support structures.

Type of 3D Structure	Lattice Constants of ZnO		Crystalite Size of ZnO for (101) Diffraction Peak, nm	Percentage Composition, %	
	a = b, nm	c, nm		ZnO	SALT
Non-impregnated sample	0.32505	0.5207	96.98	100	-
NaNO_3_ impregnated sample	0.3251	0.5208	101.19	44.94	55.06
KNO_3_ impregnated sample	0.3249	0.5206	69.95	95.55	4.45
NaNO_3_:KNO_3_ impregnated sample	0.32501	0.5206	55.64	84.45	15.55

**Table 2 membranes-12-00588-t002:** Structural parameters (calculated according to XRD data) for the PCM salts impregnated into 3D structures.

Type of 3D Structure	Impregnated PCM Phase	Lattice Constants of Salt			Average Crystalite Size, nm
		a, nm	b, nm	c, nm	
NaNO_3_ impregnated sample	NaNO_3_	0.507	0.507	1.683	µm—tens of µm
KNO_3_ impregnated sample	KNO_3_	0.642	0.541	0.917	38.00
NaNO_3_:KNO_3_ impregnated sample	NaNO_3_:KNO_3_	0.642 *	0.541 *	0.917 *	88.56 *

* KNO_3_ parameters.

**Table 3 membranes-12-00588-t003:** Results of the thermogravimetric analysis for non-impregnated 3D ZnO-CNT structure.

Type of 3D Structure	Cycle No.	Heating Cycles	Cooling Cycles
		Total Mass Loss/Gain, %	Total Mass Loss, %
Non-impregnated sample	1	−1.376	−0.245
	2	0.147	−0.229
	3	0.169	−0.243
	4	0.162	−0.241
	5	0.182	−0.241

**Table 4 membranes-12-00588-t004:** Results of the thermogravimetric analysis for PCMs impregnated 3D structures.

Type of 3D Structure	Cycle No.	Heating Cycles	Cooling Cycles
		Mass Loss/Gain, %	Mass Loss, %
NaNO_3_ impregnated sample	1	−1.6	−0.128
	2	0.104	−0.123
	3	0.121	−0.124
	4	0.116	−0.117
	5	0.126	−0.124
KNO_3_ impregnated sample	1	−3.078	−0.171
	2	0.11	−0.179
	3	0.135	−0.146
	4	0.118	−0.190
	5	0.148	−0.185
NaNO_3_:KNO_3_ impregnated sample	1	−2.098	−0.126
	2	0.061	−0.119
	3	0.145	−0.108
	4	0.086	−0.121
	5	0.082	−0.118

**Table 5 membranes-12-00588-t005:** Main peaks observed during the heating-cooling thermal cycles of the impregnated 3D structures.

Type of PCM Impregnated in 3D Structure	Cycle No.	Heating Cycles			Cooling Cycles	
		Peak1 endo, °C	Peak2 endo, °C	Peak3 endo, °C	Peak 1 exo, °C	Peak 2 exo, °C
NaNO_3_	1		277.5	310.3	295.3	
	2		276.5	306.6	295.2	
	3		276.3	304.6	295.3	
	4		276.4	304.5	295.6	
	5		276.3	304.5	295.6	
KNO_3_	1	138.4	238.8	327.6	292.6	
	2	142.2		320.4	293	
	3	141.9		319.9	293.1	
	4	141.8		320.3	292.6	
	5	141.5		320.2	293	
NaNO_3_:KNO_3_/1:1	1	137.5	249	335.8	330	306.9
	2	144.1	321.4	335.1	330.6	307.6
	3	143.7	321.2	335	330.7	307.8
	4	143.2	321	334.9	330.5	307.8
	5	142.9	320.8	334.8	330	307.7

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
