# Peer review of "Development of 3D ZnO-CNT Support Structures Impregnated with Inorganic Salts"

_membranes, 2022, doi:10.3390/membranes12060588_

Round 1

Reviewer 1 Report

In my opinion, the revised manuscript presented by the authors reaches the quality standars of this journal, therefore I recommend its publication.

Author Response

Thank you very much. 

Reviewer 2 Report

Comments

The authors prepared ZnO-CNT 3D support for impregnating NaNO3 and KNO3 as phase change materials for energy storage. The impregnated NaNO3 has a melting enthalpy of 56.09 J/g, while the NaNO3 has a very low melting enthalpy of 0.7 J/g. I don’t understand why the authors impregnated NaNO3 and KNO3 together, since the melting enthalpy also dramatically decreased comparing with that of impregnated with NaNO3. Besides, I don’t think that the thermal stability improved, because even if the two components mixed, it is hard to generate synergistic effect. Therefore, I don’t think that this manuscript is suitable for publication in Membranes. Important issues are also listed below.

  1. The logic of this study should be clarified, e.g. the relationship between impregnation with NaNO3, KNO3, and NaNO3 and KNO3.
  2. There are too many paragraphs in the Introduction section, which make the logic of this section unclear.
  3. Phrases like 'for the first time' should be avoided in the context.
  4. If the names of the substrates appear only once in the context, there is no need to define abbreviations. On the contrary, once the abbreviations defined, the full name should not appear in the following context.
  5. The authors are strongly suggested to provide the SEM images of the cross section of ZnO-CNT support before and after impregnated inorganic salts, which is very important to understand the spatial distribution of the inorganic salts.
  6. The explanation of different panels in Figure 2 should be provided.
  7. The quality of the images is very poor in the manuscript. It is hard to obtain any detailed information, especially for Figure 3b.
  8. In Line 299-302, it is hard to draw the conclusion from Figure 4 that ‘the inorganic salts impregnation started into micropores, continue with the covering of the 3D structure surface and ...’.
  9. It is important to determine that if the inorganic salts only deposited on the surface of the support or into the micro-, mesopores of the support, which is very important to understand the thermal stability of the PCMs.
  10. Since the caption of Figure 6 is the DSC-TG graph, where is the TG curve? Only DSC curves were observed.
  11. What is the CNT’s functional groups decomposed when elevating the temperature, which should be briefly mention in this study.

Reviewer 3 Report

Minor Revision needed

  1. The quality of FE-SEM images in Figure 3 need to be improved.
  2. The picture quality of all the figures are blur which should be improved.
  3. English language should be polished throughout the manuscript.
  4. TEM image of the optimized materials should be provided.
  5. Some recent papers based on carbon should be incorporated in proper places.

https://doi.org/10.1016/j.jcis.2018.03.055

https://doi.org/10.1021/acssuschemeng.1c08059

  1. XRD analysis of the synthesized materials should be provided.

Round 2

Reviewer 2 Report

Though the novelty of this manuscript is moderate, I think this manuscript can be published in Membranes since all of my questions and suggestions have been addressed.